# Expanded Newborn Screening for Inborn Errors of Metabolism at a Single Center in Louisiana (2005–2024): Outcomes

**DOI:** 10.3390/ijns11040112

**Published:** 2025-12-09

**Authors:** Jariya Upadia, Grace Noh, Kea Crivelly, Elise Aziz, Amy Cunningham, Hans C. Andersson

**Affiliations:** 1Hayward Genetics Center, Tulane University School of Medicine, 1430 Tulane Avenue, New Orleans, LA 70112, USA; gnoh@tulane.edu (G.N.); kcrivell@tulane.edu (K.C.); eaziz@tulane.edu (E.A.); acunnin@tulane.edu (A.C.); handers@tulane.edu (H.C.A.); 2Department of Pediatrics, Tulane University School of Medicine, New Orleans, LA 70112, USA

**Keywords:** newborn screening, clinical outcome, neurodevelopmental outcome, inborn error of metabolism, tandem mass spectrometry

## Abstract

This study evaluates the incidence of metabolic disorders detected from January 2005 to December 2024 and their clinical outcomes. Data were retrospectively collected from the Louisiana Newborn Screening database. Clinical outcomes were obtained through review of corresponding medical records. In addition, an electronic questionnaire assessing educational attainment and neurodevelopmental disorders was sent to the patients’ families. Of 1,230,356 infants screened, 478 were diagnosed with metabolic disorders, corresponding to an incidence of 1 in 2574 live births. The three most commonly identified conditions were biotinidase deficiency, phenylketonuria (PKU), and medium-chain acyl-CoA dehydrogenase deficiency (MCADD). During the study period, at least 11 patients died. The program demonstrated a false-positive rate of 0.93%. Twelve patients (7%) were symptomatic before or at the time of NBS result notification. Recurrent metabolic decompensations occurred in 3 of 4 maple syrup urine disease (MSUD) cases, 7 of 7 methylmalonic acidemia (MMA) cases, 1 of 4 propionic acidemia (PA) cases and 1 of 7 urea cycle defect cases. Regarding long-term outcomes, 45.7% of survey respondents reported adverse neurodevelopmental outcomes of varying severity. Early detection and timely intervention have contributed to normal or near-normal outcomes in many cases. However, the morbidity and mortality observed in some patients despite early diagnosis highlights the severity and complexity of certain metabolic conditions. Additionally, the relatively high false positive rate underscores the need for ongoing efforts to improve the specificity of screening protocols to reduce unnecessary follow-ups and mitigate potential stress for families.

## 1. Introduction

Newborn screening (NBS) enables the presymptomatic detection of infants with IEMs. It initially began with screening for a single disorder, PKU. In 1961, American microbiologist Dr. Robert Guthrie introduced a simple and cost-effective dried blood spot test using bacterial inhibition enzymes to detect elevated phenylalanine levels [1]. By the late 1960s, PKU screening had been implemented nationwide in the United States and in several other countries. Louisiana launched its newborn screening program in 1964. What began as a single-disease screening has evolved into an advanced platform capable of detecting over 50 metabolic disorders through tandem mass spectrometry (MS/MS). The application of MS/MS on dried blood spots, enabling the identification of multiple metabolites with high throughput efficiency, was first proposed in 1990 [2]. This technology was later introduced as a newborn screen pilot project in North Carolina (1997–1999), where it was used to detect fatty acid oxidation defects, organic acidemias, and amino acid disorders [3].

From 1999 to 2004, Louisiana routinely screened newborns for five disorders, as mandated by Louisiana statute §40:1299: biotinidase deficiency, galactosemia, sickle cell disease, congenital hypothyroidism, and PKU. In 2004, Louisiana began incorporating MS/MS into its NBS program. Updates to the statute expanded the screening panel to include additional metabolic disorders detectable through MS/MS technology. In 2006, the American College of Medical Genetics (ACMG) recommended a uniform panel of 29 conditions for state NBS programs [4]. Since the launch of expanded newborn screen (eNBS) in Louisiana, no formal report has been published on its screening parameters or outcomes. This paper documents the efficacy of eNBS in Louisiana from 2005 to 2024, analyzes the observed incidence of screened conditions, and provides a review of patient outcomes. Despite the widespread implementation of NBS programs, there is a relative paucity of studies reporting on long-term clinical outcomes. This scarcity of data underscores the need for further studies, including the present work, to better understand the real-world impact of NBS on patient health and development.

## 2. Material and Methods

### 2.1. Study Design

This was a retrospective study of newborn screening data collected between January 2005 and December 2024. Data from the state’s Microsoft Access database, used for tracking follow-up, were exported into a Microsoft Excel file. Confirmatory results from electronic and paper medical records were reviewed.

### 2.2. Screening Panel

The NBS panel detects a range of metabolic disorders using MS/MS, including fatty acid oxidation disorders, organic acidemias, urea cycle defects, and amino acidopathies. In addition to MS/MS-detected conditions, the NBS program also screens for biotinidase deficiency, galactosemia, hemoglobinopathies, severe combined immunodeficiency (SCID), congenital hypothyroidism, congenital adrenal hyperplasia (CAH), spinal muscular atrophy (SMA), mucopolysaccharidosis type I (MPS I), and Pompe disease. However, this report does not include data on hemoglobinopathies, SCID, SMA, MPS I, Pompe disease, CAH or congenital hypothyroidism.

### 2.3. Data Collection

Data on total births, the number of infants with presumptive positive newborn screens, and confirmed cases of IEMs were collected. Diagnoses were confirmed by the treating medical geneticist based on clinical history, physical examination findings, and biochemical and/or molecular testing. Recommended confirmatory tests for each screened condition were based on the ACMG ACT Sheets and Algorithms. Patient demographic information (age, sex), diagnoses, treatments, and clinical outcomes were obtained from existing medical records, including both paper and electronic sources. Clinical outcomes including morbidity, mortality and developmental status were specifically analyzed for patients with IEMs detected by MS/MS. Certain metabolic conditions were excluded from this analysis: biotinidase deficiency, galactosemia, and carnitine uptake defect (CUD) due to incomplete medical records or because patients with these conditions were not followed at our clinic. Additionally, some conditions typically associated with a mild clinical course were not included in this analysis including 3-methylcrotonyl-CoA carboxylase (3-MCC) deficiency, isobutyryl-CoA dehydrogenase deficiency (IBDD), 2-methylbutyryl-CoA dehydrogenase deficiency (2-MBDD), and short-chain acyl-CoA dehydrogenase deficiency (SCADD). Patients with hemoglobinopathies, SCID, SMA, CAH, and congenital hypothyroidism were not included in this analysis because our metabolic center does not follow patients with these conditions, and therefore outcome data were not available. Similarly, Pompe disease and MPS I were not included, as outcomes for these disorders have been extensively reported by multiple centers, and incorporating them here would require additional data collection beyond the scope of this manuscript.

The screened conditions were categorized into three groups based on disease severity. First-tier conditions include those with the potential for acute presentation or metabolic decompensation: tyrosinemia type I (HT1), maple syrup urine disease (MSUD), citrullinemia type I, argininosuccinic aciduria (ASA), glutaric acidemia type I (GA1), isovaleric acidemia (IVA), methylmalonic acidemia (MMA), propionic acidemia (PA), holocarboxylase synthetase deficiency (HLCSD), MCADD, very long-chain acyl-CoA dehydrogenase deficiency (VLCADD), multiple acyl-CoA dehydrogenase deficiency (MADD), carnitine palmitoyltransferase II (CPT2) deficiency, and carnitine-acylcarnitine translocase (CACT) deficiency. Second-tier conditions, which are typically associated with a chronic disease course, include PKU, homocystinuria (HCU) caused by cystathionine beta-synthase deficiency, 6-pyruvoyl-tetrahydropterin synthase (PTPS) deficiency, cobalamin C deficiency, biotinidase deficiency, and classic galactosemia. Third-tier conditions, considered mild or benign, include methionine adenosyltransferase I (MAT I) deficiency, 3-MCC deficiency, IBDD, MBDD, and SCADD.

### 2.4. Questionnaire

An anonymous electronic questionnaire was distributed via email to patients or their guardians, using the most recent contact information available in the database. Patients with galactosemia, biotinidase deficiency, and third-tier diseases were excluded from the study, as our center did not provide ongoing care for these conditions and contact information for these patients was not available. The survey included questions on current age, school grade level, receipt of special education services or accommodations, and presence of neurodevelopmental or psychiatric conditions such as autism spectrum disorder (ASD), epilepsy, developmental delay, speech delay, learning disability, attention-deficit/hyperactivity disorder (ADHD), and other behavioral or psychiatric diagnoses. The survey was voluntary, and responses were collected securely through Research Electronic Data Capture (REDCap) over an 8-week period with weekly reminders sent to non-respondents. This timeframe was chosen because it provides sufficient opportunity for participants to respond, maximizes response rates without causing survey fatigue, and balances timely data collection with study efficiency. Eight weeks is consistent with common practice for clinical and academic outcome surveys of similar scope. Descriptive statistics were used to summarize responses.

### 2.5. Participant Information Precautions and IRB

All procedures were reviewed and approved by the Tulane Institutional Review Board.

## 3. Results

### 3.1. Incidence

A total of 1,230,356 newborns were screened in Louisiana between January 2005 and December 2024. Confirmatory metabolic and/or molecular testing identified 478 cases of IEMs, corresponding to an overall incidence of 1 in 2574. Excluding galactosemia and biotinidase deficiency, 11,894 infants screened positive, of whom 342 were confirmed to have a metabolic disorder. This corresponded to an incidence of 1 in 3598 and a false-positive rate of 0.93% (Table 1). The most commonly detected conditions were biotinidase deficiency (1 in 10,101), PKU (1 in 13,986), MCADD (1 in 16,667), SCADD (1 in 27,932) and VLCADD (1 in 40,983).

Among the diagnosed cases, 185 infants had disorders classified as first-tier conditions. Most common first-tier disorders were MCADD, VLCADD, and GA1. An additional 231 infants were diagnosed with metabolic disorders associated with chronic disease course (second-tier), with biotinidase deficiency and PKU being the most prevalent. Sixty-two infants were diagnosed with metabolic conditions considered mild or benign, with SCADD being the most frequent.

### 3.2. Positive Predictive Value (PPV)

The overall PPV of the NBS program was 4%. Among all screened conditions, PKU demonstrated the highest PPV at 26.1%. PPVs for other disorders were as follows: SCADD, 13.1%; VLCADD, 12.2%; MCADD, 12.0%; and GA1, 10.2%. IVA; MSUD and HCU each had PPVs of less than 1%.

### 3.3. Outcomes of Each Metabolic Disease (Appendix A)

Among cases diagnosed with first-tier metabolic diseases with potential for acute presentation or metabolic decompensation (excluding galactosemia), twelve cases (7%) were symptomatic prior to or at the time the NBS results became available. Recurrent episodes of metabolic decompensation were observed in 3 of 4 MSUD cases, 7 of 7 MMA cases, 1 of 4 PA cases, and 1 of 5 MADD cases. Liver transplantation was performed in 2 of 4 MSUD cases, while combined liver and kidney transplantation was performed in 4 of 7 MMA cases. Metabolic stroke occurred in 2 of 18 GA1 cases and 2 of 7 MMA cases. Developmental delay or intellectual disability was common, affecting 2 of 4 MSUD cases, 10 of 18 GA1 cases, 6 of 7 MMA cases, and 2 of 4 PA cases. Among first-tier diseases, mild variants were observed, including all 9 cases of IVA and 3 of 5 cases of citrullinemia type 1. None of the patients with mild variants experienced metabolic decompensation.

Among second-tier metabolic diseases, which are typically associated with a chronic disease course, 2 of 3 HCU cases achieved good metabolic control. Developmental delay was observed in 2 of 3 HCU cases and in all 5 cases of cobalamin C deficiency. Clinical outcomes for biotinidase deficiency, and CUD were not available in this study.

### 3.4. Deaths

Among patients diagnosed with first-tier conditions, eleven deaths were recorded. The underlying diagnoses were MCADD (3), VLCADD (1), MADD (2), long-chain 3-hydroxyacyl-CoA dehydrogenase deficiency (LCHAD) (1), carnitine palmitoyltransferase II (CPT2) deficiency (2), carnitine-acylcarnitine translocase (CACT) deficiency (1), and HLCSD (1) (Table 2).

### 3.5. Survey Results

Of the 221 patients invited to participate in the survey, 81 responded (37%). Among the respondents, 27 (33.3%) had PKU, 24 (29.6%) had MCADD, 10 (12.3%) had VLCADD, 4 (5.0%) had MMA due to methylmalonyl-CoA mutase deficiency (MMA-mut), 3 (3.7%) had ASA, 2 (2.5%) each had Cobalamin C deficiency, Citrullinemia type 1, Homocystinuria, or GA1, and 1 (1.2%) each had LCHADD, IVA, PA, HT1 1, or MMA caused by cobalamin B deficiency (MMA-CblB).

Of the total 81 respondents, 64 were attending school. Among these 64 patients, 4 (4.9%) were in preschool (Pre-K, ages 4–5), 23 (28.4%) were in elementary school (kindergarten through 5th grade), 13 (16.0%) were in middle school (6th through 8th grade), 16 (19.8%) were in high school (9th through 12th grade), 3 (3.7%) had graduated from high school or obtained a GED, 4 (4.9%) were currently in college, and 1 (1.2%) had not completed high school (left in 11th grade). Among the 56 patients enrolled in elementary through high school, 39 (70%) were performing at grade level. Of the 17 patients who were below grade level, diagnoses included MMA-mut (*n* = 4), cobalamin C deficiency (*n* = 3), ASA (*n* = 3), PKU (*n* = 2), MCADD (*n* = 2), and one each with GA1, MMA-CblB, and PA.

Of the 81 respondents, 37 patients (45.7%) were diagnosed with at least one neurodevelopmental or neuropsychiatric condition, including autism, seizures, developmental delay, learning disability, ADHD, anxiety, or depression. Among these, 22 (27.2%) had developmental delay, 15 (18.5%) a learning disability, 14 (17.3%) ADHD, 11 (13.6%) autism, 6 (7.4%) epilepsy, 6 (7.4%) anxiety, and 4 (4.9%) depression. By diagnosis, at least one condition was present in 9 of 27 patients with PKU, 5 of 24 with MCADD, 5 of 10 with VLCADD, all 4 patients with MMA, all 3 with ASA, 2 of 2 for each of Cobalamin C deficiency, homocystinuria, and GA1, and 1 of 1 for each of MMA-CblB, LCHAD, IVA, PA, and HT1.

In addition, 39 patients (48.1%) received at least one special service, including Individualized Education Plans (IEP), Early Steps, or speech, physical, or occupational therapy. Among these, 29 (35.8%) had an IEP at school, 21 (25.9%) participated in Early Steps programs, 28 (34.6%) received speech therapy, 16 (19.8%) received physical therapy, and 22 (27.2%) received occupational therapy. By diagnosis, 9 of 27 patients with PKU, 6 of 24 with MCADD, 6 of 10 with VLCADD, all 4 patients with MMA, all 3 with ASA, and both patients with Cobalamin C deficiency and GA1 required at least one special service.

## 4. Discussion

The overall incidence of amino acid disorders, organic acid disorders, and fatty acid oxidation disorders identified in Louisiana was 1 in 3598, compared to an incidence of 1 in 4300 for the same subset of conditions in North Carolina [3]. When compared to California, the incidence was similar: 1 in 6500 in California and 1 in 6152 in Louisiana for a comparable subset of conditions [5]. When compared to the 10-year incidence rates in the United States reported by the National Newborn Screening Information System (NNSIS) [6], the incidence rates of MCADD, PA, IVA, and homocystinuria in Louisiana were similar. However, the incidence rates for HT1, VLCADD, GA1, and PKU in Louisiana were approximately 3.2, 1.5, 1.4, and 1.2 times higher, respectively, than those reported in the NNSIS dataset. Several very rare conditions, including CACT deficiency, HLCSD, LCHADD, methylmalonic acidemia (CblB type), PTPS deficiency, and MAT I deficiency, were also identified in Louisiana. Notably, argininemia, HMG-CoA lyase def, Beta-ketothiolase def, ethylmalonic encephalopathy, tyrosinemia type II and CPT I deficiency were not detected during the 19-year study period. It is important to note that these conditions are extremely rare, and their absence in our cohort is consistent with their low expected incidence in the United States [7,8,9,10].

PPVs varied widely across conditions, ranging from less than 1% to 26%. Disorders with lower PPVs resulted from a higher number of cases requiring confirmatory testing. The overall false-positive rate among screened newborns for IEMs (excluding galactosemia and biotinidase deficiency) was 0.93%, which is substantially higher than rates reported by the California NBS program and the Mayo Clinic. In practical terms, this corresponds to approximately 1 true positive case in every 103 infants receiving a positive screening result. Notably, the California NBS program reduced its false-positive rate to 0.07% by adjusting cutoff values, while the Mayo Clinic achieved a rate of 0.09% through the implementation of second-tier testing for selected conditions [5,11]. A higher false-positive rate increases the burden of unnecessary confirmatory testing and may heighten parental stress and anxiety, highlighting the importance of optimizing screening algorithms through the use of second-tier tests or adjustment of laboratory cutoff values to improve specificity.

NBS enables early diagnosis and prompt treatment of metabolic disorders, helping to prevent metabolic decompensation, chronic complications, and death. However, not all cases of screened conditions are detected by NBS [12,13,14]. To the best of our knowledge, five cases were missed by the Louisiana NBS program during the study period. These included two siblings with the myopathic form of CPT II deficiency, one individual with ASA, one with citrin deficiency, and one with cobalamin C deficiency (Appendix A). Despite early diagnosis through NBS, 11 out of 185 cases diagnosed with first-tier conditions resulted in death, attributable to the severity of the underlying diseases (5 cases), mismanagement by healthcare providers unfamiliar with these conditions (2 case), non-metabolic causes (2 cases), and unknown causes (2 cases). Although newborn screening programs follow standardized procedures, there are opportunities to reduce reporting time for high-risk conditions. Strategies such as prioritizing critical results, streamlining laboratory workflows, and enhancing electronic reporting could allow earlier identification and intervention, which may improve clinical outcomes.

Outcomes vary depending on the specific disorder. Although early diagnosis through NBS and appropriate treatment can significantly improve outcomes, neurological damage and episodes of metabolic decompensation are not always preventable [15,16,17,18]. In our cohort, two of 18 patients with GA1 and two of six patients with MMA-mut experienced metabolic strokes following episodes of acute metabolic decompensation. Newborn screening did not completely prevent metabolic decompensation, since many patients with disorders predisposed to these events experienced them despite early diagnosis. A normal neurodevelopmental outcome is expected in patients with treatable conditions such as HT1 and biotinidase deficiency, as well as in conditions where adherence to standard management protocols can prevent adverse neurological outcomes, such as MCADD and VLCADD. Although there is no definitive cure for disorders such as PKU, MSUD, MMA, PA, MADD, HCU, and PTPS deficiency, adherence to treatment can reduce long-term complications. NBS enables the detection of milder variants of IEMs. In our cohort, most patients diagnosed with IVA, VLCADD, and citrullinemia type I were either asymptomatic or presented with mild forms of disease, and no severe complications were observed.

The majority of adolescents with metabolic diseases diagnosed through NBS demonstrated normal cognitive and academic outcomes, with most attending regular schools [18]. However, about half of respondents in this study were diagnosed with at least one neurodevelopmental or neuropsychiatric condition requiring at least one special service, including IEP, Early Steps, or speech, physical, or occupational therapy. This finding suggests that while NBS facilitates early diagnosis and intervention, a significant proportion of patients remain at risk for adverse neurodevelopmental outcomes. Possible explanations include incomplete metabolic control despite treatment, variability in disease severity and individual response, and the cumulative impact of acute decompensation. These factors highlight the importance of ongoing surveillance, continual optimization of treatment strategies, and the provision of supportive educational and therapeutic services to enhance long-term outcomes. It is important to recognize that neurodevelopmental and neuropsychiatric conditions have a multifactorial origin and background incidence of developmental delay is roughly 4%. In addition to complications directly related to the metabolic disorder, other contributing factors such as genetic predispositions and environmental influences also play significant roles [19,20,21]. Notably, approximately 70% of the respondents performed well academically, a finding that underscores the profound impact of newborn screening. Many of the metabolic disorders included in this study carry a high risk of neurodevelopmental complications if not detected and managed early. Without timely diagnosis through NBS and the subsequent initiation of treatment, a substantially higher number of children would not achieve academic success.

Several limitations should be considered when interpreting the findings of this study. First, the retrospective design relies on the completeness and accuracy of historical medical records and state databases, which may have been affected by data losses and disruptions, particularly in the early years following Hurricane Katrina. Second, the survey component is subject to responder bias, as only 81 of 221 invited participants responded, potentially overrepresenting patients with better outcomes or higher engagement with care. This may limit the generalizability of the neurodevelopmental and educational outcome data. The survey was distributed via email to addresses available in the medical record, with weekly reminders over an 8-week period. Despite these efforts, some patients may not have read the emails or messages may have been routed to spam/junk folders, which could have contributed to the lower participation. Factors such as the method of distribution, email validity, and patient engagement likely influenced the response rate. Third, certain metabolic conditions, including galactosemia, biotinidase deficiency, and SCADD, were excluded from the clinical outcome survey and outcome analyses due to incomplete follow-up or unavailable contact information, which could skew estimates of overall outcomes. Fourth, several cases were lost to follow-up, which is a recognized challenge in newborn screening programs. While the exact reasons for loss to follow-up are not always known, factors such as socioeconomic status, geographic barriers, and patient mobility may play a role. To address this limitation and improve retention, potential strategies include enhanced patient tracking, use of electronic reminders, and community outreach programs to engage families. Finally, outcomes reported in this study may be influenced by unmeasured genetic, environmental, and socioeconomic factors, which were not systematically captured, further emphasizing the multifactorial nature of neurodevelopmental and academic performance among affected individuals.

## 5. Conclusions

NBS remains a valuable tool for early detection and intervention, significantly improving outcomes by preventing severe metabolic crises and enabling many affected children, approximately 70 percent in our cohort, to achieve good academic performance. Despite early diagnosis and treatment, a substantial proportion of patients still experience neurodevelopmental or neuropsychiatric complications, reflecting the multifactorial nature of these conditions influenced by metabolic control, disease severity, as well as genetic and environmental factors. Deaths are significantly reduced but not completely eliminated, especially in patients with co-morbidities or presenting with severe sequelae at birth. The relatively high false-positive rate emphasizes the need to refine screening protocols to reduce unnecessary follow-ups and associated stress.

## Figures and Tables

**Table 1 IJNS-11-00112-t001:** Incidence of metabolic diseases identified through newborn screening in Louisiana from January 2005 to December 2024, compared to the incidence of these disorders from NNSIS database from 2001 to 2011.

Disease	Number of Cases in Louisiana 2005–2024 (n)	Louisiana Incidence 2005–2024 1:X	Ten Years of Incidence Data from NNSIS (2001–2011) * 1:X
First tier (Acute)			
Argininosuccinic aciduria (ASA)	3	410,119	305,032
Carnitine Palmitoyl Transferase II (CPT 2) deficiency	4	307,589	
Carnitine-acylcarnitine translocase (CACT) deficiency	1	1,230,356	
Citrullinemia type I	5	246,071	155,679
Galactosemia—classical	14	102,530	53,554
Glutaric Acidemia type I (GA1)	18	68,353	92,302
Holocarboxylase synthetase deficiency (HLCSD)	1	1,230,356	1,927,913
Isovaleric Acidemia (IVA)	9	136,706	159,150
Long-chain 3-hydroxyacyl-coenzyme A dehydrogenase deficiency (LCHADD)	1	1,230,356	363,738
Maple Syrup Urine Disease (MSUD)	4	307,589	197,714
Medium Chain Acyl-CoA Dehydrogenase Deficiency (MCADD)	74	16,626	17,759
Methylmalonic Acidemia (MMA)-mutase deficiency	6	205,059	159,614
Methylmalonic Acidemia (MMA) CblA,B	1	1,230,356	410,343
Multiple acyl-CoA dehydrogenase deficiency (MADD)	5	246,071	
Propionic Acidemia (PA)	4	307,589	238,346
Tyrosinemia Type I (HT1)	5	246,071	781,144
Very long-chain acyl-CoA dehydrogenase deficiency (VLCADD)	30	41,012	63,481
Total	185	6650	
Second tier (Subacute/Chronic)			
Biotinidase deficiency—partial and profound	122	10,085	67,766 Partial 24,957
Carnitine Uptake Deficiency (CUD)	12	102,530	142,236
Cobalamin C deficiency	5	246,071	
Homocystinuria (HCU)	3	410,119	456,726
Phenylketonuria (PKU)	88	13,981	16,500
6-pyruvoyl-tetrahydropterin synthase (PTPS) deficiency	1	1,230,356	
Total	231	5326	
Third tier (Asymptomatic/mild variant)			
Isobutyryl-CoA dehydrogenase deficiency (IBDD)	3	410,119	
Methionine adenosyltransferase I (MAT I) deficiency	1	1,230,356	
2-methylbutyryl-CoA dehydrogenase deficiency (2-MBDD)	1	1,230,356	
3-methylcrotonyl-CoA carboxylase (3-MCC) deficiency	13	94,643	38,636
Short-chain acyl-CoA dehydrogenase deficiency (SCADD)	44	27,962	
Total	62	19,844	

* NNSIS; National Newborn Screening Information System.

**Table 2 IJNS-11-00112-t002:** Diagnoses and Causes of Death Among 11 Patients with Metabolic Disorders.

Diagnosis (N)	Cause of Death
MCADD (3)	Case 1: The patient died on the first day of life due to fetal hydrops with severely hypoplastic lungs. Case 2: The patient, a sibling of Case 1, died on the third day of life of unknown cause. NBS results became available on day 5 of life. Case 3: The patient died of unknown cause at an unspecified age. The diagnosis of MCADD was made via NBS. The patient was lost to follow-up after 3 months of age.
VLCADD (1)	The patient died at 13 months of age during an intercurrent illness, having received partial or incomplete treatment that did not fully follow the recommended protocol.
LCHADD (1)	The patient died from an unknown cause at 8 days of life, on the same day the NBS result became available
CPT2 def (2)	Case 1: NBS result became available on day 10 of life; the patient died in the NICU at 31 days of age due to metabolic decompensation, compounded by mismanagement of the condition. Case 2: The patient died at age 14 years from complications of COVID-19. Prior to his death, he had been clinically stable with effective management of CPT II deficiency.
CACT def (1)	The patient was started on a high-carbohydrate, low long-chain fat diet, supplemented with MCT oil and carnitine. Despite these interventions, she developed dilated cardiomyopathy with progressively worsening cardiac function. The patient passed away at 22 months of age due to cardiac failure.
MADD (2)	Case 1: The patient died in the first week of life before NBS results were available. Although medical records are unavailable, urine organic acid analysis on day 2 of life strongly suggested MADD. No molecular testing was performed. This case was likely severe neonatal-onset MADD. Case 2: The patient died at 2 years following recurrent metabolic decompensation, dilated cardiomyopathy, and congestive heart failure.
HLCSD (1)	The patient was admitted to the NICU on day 1 of life for hypothermia, seizures, and worsening metabolic acidosis. High-dose biotin (30 mg/day) and carnitine supplementation were initiated upon diagnosis. At 21 months of age, the patient died from intractable metabolic acidosis triggered by a GI infection, despite intensive treatment including continuous renal replacement therapy (CRRT).

## Data Availability

The data presented in this study are available on request from the corresponding author because they consist of raw data and are not self-explanatory.

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
