# Peer review of "Expanded Newborn Screening for Inborn Errors of Metabolism at a Single Center in Louisiana (2005–2024): Outcomes"

_2409-515X, 2025, doi:10.3390/ijns11040112_

Round 1

Reviewer 1 Report

Comments and Suggestions for Authors

This manuscript is about the experience of the Louisiana newborn screening. The length of time described in the study is very comprehensive, and the manuscript is well written. I do think that some aspects should be clarified or further reviewed. I do think that additional of discussion and a few clarifications are required.

a)  Why were hemoglobinopathies, SCID, SMA, MPS I, Pompe disease, CAH or congenital hypothyroidism excluded from this manuscript? I think they are extremely relevant, and it would greatly help the readers to learn the experience of Louisiana with these conditions. I understand that several of these were implemented more recently, but any experience shared would helper other centers.

b) Page 2, lines 85-87; for the patients that are not followed at your center, does the screening lab get more information? How is the follow-up achieved in a consistent manner throughout the state?

c) For the questionnaire why were only 8 weeks used for data collection? It does feel like quite a short amount of time considering how massive the whole study and follow-up were.

d) For a lot of the cases identified it took 5 to 8 days for the results to be reported, and for some patients that was too late. Is there a reason why it is taking so long for the results? Based on the clinical outcomes and deaths, is there anything that the program can do to improve and have results quicker?

e) The response rate for the questionnaire was very low, can the authors discuss more about this? Some other programs have reached much higher rates. Was it lack of funding and personnel impacting the response rates? Please discuss more about this in the paper.

f) Several cases seemed to be lost to follow-up. I understand that this is a challenge in several other programs. Can the authors discuss more about this and propose solutions to have lower loss of follow-up rates?

Overall this is a very nice study, it has impressive body of data. But I feel like that the limitations of the study should be discussed more, and that all disorders screened by the program should be included to reflect the whole program. 

The study also does not dive too much about second-tier and confirmatory testing. The addition of this information would be very helpful for the readers.

Author Response

Thank you very much for taking the time to review our manuscript, “Expanded Newborn Screening for Inborn Errors of Metabolism in Louisiana (2005–2024): Outcomes.” We have carefully considered the reviewers’ comments and greatly appreciate their time, effort, and the many helpful suggestions and thoughtful critiques.

Below, we have addressed each comment individually. The original comment is italicized, our response follows “>>>”, and any text added to the manuscript is highlighted in red.

Reviewer 1

This manuscript is about the experience of the Louisiana newborn screening. The length of time described in the study is very comprehensive, and the manuscript is well written. I do think that some aspects should be clarified or further reviewed. I do think that additional discussion and a few clarifications are required.

  1. a) Why were hemoglobinopathies, SCID, SMA, MPS I, Pompe disease, CAH or congenital hypothyroidism excluded from this manuscript? I think they are extremely relevant, and it would greatly help the readers to learn the experience of Louisiana with these conditions. I understand that several of these were implemented more recently, but any experience shared would helper other centers.

>>> Thank you for this insightful comment. The scope of our manuscript was specifically focused on inborn errors of metabolism detected by tandem mass spectrometry (MS/MS). Inclusion of hemoglobinopathies, SCID, SMA, CAH, and congenital hypothyroidism was not feasible because our metabolic center does not follow patients with these conditions, so we do not have outcome data to report.

Regarding Pompe disease and MPS I, outcomes have already been reported extensively by multiple centers, and including them here would require additional data collection that is beyond the scope of this manuscript. We have clarified this limitation in the Methods section of the manuscript to ensure readers understand that the focus is on MS/MS-detectable metabolic disorders.

We provided the reason for excluding this disease on Page 2, line 85-88

Certain metabolic conditions were excluded from this analysis: biotinidase deficiency, galactosemia, and carnitine uptake defect (CUD) due to incomplete medical records or because patients with these conditions were not followed at our clinic.

and Page 3, line 94-99

Patients with hemoglobinopathies, SCID, SMA, CAH, and congenital hypothyroidism were not included in this analysis because our metabolic center does not follow patients with these conditions, and therefore outcome data were not available. Similarly, Pompe disease and MPS I were not included, as outcomes for these disorders have been extensively reported by multiple centers, and incorporating them here would require additional data collection beyond the scope of this manuscript.”

>>> To clarify the scope of our data, we also revised the title to specify that the results reflect a single metabolic center in Louisiana.

  1. b) Page 2, lines 85-87; for the patients that are not followed at your center, does the screening lab get more information? How is the follow-up achieved in a consistent manner throughout the state?

>>> Thank you for your comment. Patients who are not followed at our center are managed by another metabolic center in the area. All centers follow the standardized protocols recommended by the American College of Medical Genetics and Genomics (ACMG). The state newborn screening program communicates results to the appropriate metabolic centers, ensuring that follow-up is performed in a consistent and standardized manner throughout Louisiana.

>>> We added “. Diagnoses were confirmed by the treating medical geneticist based on clinical history, physical examination findings, and biochemical and/or molecular testing. Recommended confirmatory tests for each screened condition were based on the ACMG ACT Sheets and Algorithms.” Page 2, line 80-83

  1. c) For the questionnaire why were only 8 weeks used for data collection? It does feel like quite a short amount of time considering how massive the whole study and follow-up were.

>>> Thank you for this comment. The questionnaire was distributed to patients via REDCap over an 8-week period, with weekly reminders sent to non-respondents. We believe this period was sufficient to allow patients to respond while maintaining timely data collection, as weekly reminders helped maximize participation and ensure a meaningful response rate.

We have added clarification in the Methods section to explain the rationale for the 8-week data collection period, page 3, line 125-129

The survey was voluntary, and responses were collected securely through Research Electronic Data Capture (REDCap) over an 8-week period with weekly reminders sent to non-respondents. This timeframe was chosen because it provides sufficient opportunity for participants to respond, maximizes response rates without causing survey fatigue, and balances timely data collection with study efficiency. Eight weeks is consistent with common practice for clinical and academic outcome surveys of similar scope.

  1. d) For a lot of the cases identified it took 5 to 8 days for the results to be reported, and for some patients that was too late. Is there a reason why it is taking so long for the results? Based on the clinical outcomes and deaths, is there anything that the program can do to improve and have results quicker?

>>> Thank you for this important comment. The turnaround time for newborn screening results in Louisiana is typically 3–6 days after sample collection. Since samples are usually collected at 24–48 hours of life, the majority of patients receive results around day 4–8 of life. This timeline reflects current workflow, including specimen transport, laboratory processing, and confirmatory testing, and is generally consistent with national standards. While earlier reporting could potentially improve outcomes in some cases, efforts such as prioritizing high-risk results or streamlining laboratory processes may help further reduce turnaround time. We have added this clarification to the manuscript, page 7, line 256-260.

Although newborn screening programs follow standardized procedures, there are opportunities to reduce reporting time for high-risk conditions. Strategies such as prioritizing critical results, streamlining laboratory workflows, and enhancing electronic reporting could allow earlier identification and intervention, which may improve clinical outcomes.”

  1. e) The response rate for the questionnaire was very low, can the authors discuss more about this? Some other programs have reached much higher rates. Was it lack of funding and personnel impacting the response rates? Please discuss more about this in the paper.

>>> Thank you for this comment. We acknowledge that the response rate for our questionnaire was 37%, which is lower than some other programs. The survey was sent via email to addresses available in the medical record, with weekly reminders over an 8-week period. Despite these efforts, some patients may not have seen or read the emails, or messages may have been routed to spam/junk folders, which could have affected participation. We have added a discussion in the manuscript to acknowledge this limitation and note that factors such as method of distribution, email validity, and patient engagement may have contributed to the lower response rate, page 8, line 306-310.

The survey was distributed via email to addresses available in the medical record, with weekly reminders over an 8-week period. Despite these efforts, some patients may not have read the emails or messages may have been routed to spam/junk folders, which could have contributed to the lower participation. Factors such as the method of distribution, email validity, and patient engagement likely influenced the response rate.”

  1. f) Several cases seemed to be lost to follow-up. I understand that this is a challenge in several other programs. Can the authors discuss more about this and propose solutions to have lower loss of follow-up rates?

>>> Thank you for this important comment. Loss to follow-up is a recognized challenge in newborn screening programs, and several cases in our cohort were indeed not fully followed. While the exact reasons are not always known, factors such as socioeconomic status, geographic barriers, and patient mobility may contribute, as Louisiana has a relatively high proportion of patients from underserved communities. We have added a discussion in the manuscript acknowledging this limitation and highlighting potential strategies to reduce loss to follow-up, page 8, line 314-319.

Fourth, several cases were lost to follow-up, which is a recognized challenge in newborn screening programs. While the exact reasons for loss to follow-up are not always known, factors such as socioeconomic status, geographic barriers, and patient mobility may play a role. To address this limitation and improve retention, potential strategies include enhanced patient tracking, use of electronic reminders, and community outreach programs to engage families.”

Overall this is a very nice study, it has impressive body of data. But I feel like that the limitations of the study should be discussed more, and that all disorders screened by the program should be included to reflect the whole program. 

>>> Thank you for your thoughtful feedback and kind words. We agree that discussing limitations is important. We have added a section acknowledging that some disorders were followed at another metabolic center, which limited the completeness of outcome data for those conditions. Additionally, Pompe disease and MPS I were only recently added to the screening panel in 2022, and cut-off values are still being refined, which we have noted as a limitation. To clarify the scope of our data, we also revised the title to specify that the results reflect a single metabolic center in Louisiana.

The study also does not dive too much about second-tier and confirmatory testing. The addition of this information would be very helpful for the readers.

>>> Thank you for your suggestion. In this study, we focused specifically on disease outcomes, rather than detailing second-tier or confirmatory testing. For context, all diagnoses were confirmed by the treating medical geneticist based on clinical history, physical examination findings, and biochemical and/or molecular testing, with recommended confirmatory tests for each screened condition following the ACMG ACT Sheets and Algorithms. We have added this statement to clarify our approach.

Page 2, lines 80-83

Diagnoses were confirmed by the treating medical geneticist based on clinical history, physical examination findings, and biochemical and/or molecular testing. Recommended confirmatory tests for each screened condition were based on the ACMG ACT Sheets and Algorithms.”

Reviewer 2 Report

Comments and Suggestions for Authors

The authors report on the incidence of metabolic disorders detected by regional universal newborn screening of 1,230,356 newborns between 2005 and December 2024 and on clinical outcomes of true positives. They report a false-positive rate of 0.93% and an overall PPV of 4%. Twelve patients (7%) were symptomatic before or at the time of NBS result notification. Of the 221 patients invited to participate in a survey of clinical outcomes, 81 responded (37%) and almost half of these had at least one neurodevelopmental or psychiatric condition.
The authors conclude that the relatively high false positive rate underscores the need for ongoing efforts to improve the specificity of screening protocols to reduce unnecessary follow-ups and mitigate potential stress for families and that the high rate of morbidity reflects the multifactorial nature of these conditions influenced by metabolic control, disease severity, as well as genetic and environmental factors.

This is a very well written paper with a clear message. It contributes valuable data to the discussion around expanded universal newborn screening for metabolic disorders, especially since the authors have obtained some clinical outcome data, which is rare.

Criticisms

  • It would help with the interpretation of the data if the authors could add a table with the screening parameters and with the cut-off values that were used for screening.
  • The formatting of Table 1 appears inconsistent in the draft.
  • The sentence “Established treatment protocols were not followed, and the patient was taken home against medical advice, resulting in death at 13 months of age.” in Table 2 should perhaps be shortened or re-phrased to avoid unnecessary conflict. This is a very rare condition and the patient may be identifiable.

Author Response

Thank you very much for taking the time to review our manuscript, “Expanded Newborn Screening for Inborn Errors of Metabolism in Louisiana (2005–2024): Outcomes.” We have carefully considered the reviewers’ comments and greatly appreciate their time, effort, and the many helpful suggestions and thoughtful critiques.

Below, we have addressed each comment individually. The original comment is italicized, our response follows “>>>”, and any text added to the manuscript is highlighted in red.

Please note that the current manuscript length is 11 pages, including 2 tables.

We look forward to your feedback and further guidance.

Reviewer 2

The authors report on the incidence of metabolic disorders detected by regional universal newborn screening of 1,230,356 newborns between 2005 and December 2024 and on clinical outcomes of true positives. They report a false-positive rate of 0.93% and an overall PPV of 4%. Twelve patients (7%) were symptomatic before or at the time of NBS result notification. Of the 221 patients invited to participate in a survey of clinical outcomes, 81 responded (37%) and almost half of these had at least one neurodevelopmental or psychiatric condition.

The authors conclude that the relatively high false positive rate underscores the need for ongoing efforts to improve the specificity of screening protocols to reduce unnecessary follow-ups and mitigate potential stress for families and that the high rate of morbidity reflects the multifactorial nature of these conditions influenced by metabolic control, disease severity, as well as genetic and environmental factors.

This is a very well written paper with a clear message. It contributes valuable data to the discussion around expanded universal newborn screening for metabolic disorders, especially since the authors have obtained some clinical outcome data, which is rare.

Criticisms

  1. It would help with the interpretation of the data if the authors could add a table with the screening parameters and with the cut-off values that were used for screening.

>>> Thank you for the suggestion. Our study focuses on the long-term outcomes of follow-up cases and was not designed to systematically evaluate screening cut-off thresholds. Moreover, these cut-offs have changed over time for many conditions, and including them would not meaningfully aid in interpreting the relationship between presumptive positives and outcomes in this context.

  1. The formatting of Table 1 appears inconsistent in the draft.

>>>Thank you for pointing this out. The formatting of Table 1 in the original manuscript was consistent; however, some inconsistencies appeared when the table was generated by the journal submission portal. We have reviewed and corrected the formatting to ensure consistency and clarity in the revised submission.

  1. The sentence “Established treatment protocols were not followed, and the patient was taken home against medical advice, resulting in death at 13 months of age.” in Table 2 should perhaps be shortened or re-phrased to avoid unnecessary conflict. This is a very rare condition and the patient may be identifiable.

      >>> We agreed with this suggestion, therefore, we revised this sentences to “The patient died at 13 months of age during an intercurrent illness, having received partial or incomplete treatment that did not fully follow the recommended protocol.”, page 5, table 2

Reviewer 3 Report

Comments and Suggestions for Authors

This work provides valuable information on the results of newborn screening in a specific location.

Abstract is OK

Keywords: words associated with the text should be included; suggestions include: neurodevelopmental outcomes, early onset, MCADD, newborn screening...

Introduction is OK

Materials and Methods

Line 75: should explain why these pathologies are not included

Line 84: add the phrase: "Certain metabolic conditions were excluded from this analysis: biotinidase..."

Results

Line 177: should specify that of the total cases (81), 64 attended school.

Discussion

Line 215: These pathologies that were not found after 19 years. Are they common in the USA, or is it a genetically distinct population? This should be discussed, as it is striking that so many pathologies have gone undiagnosed after 19 years.

Line 260: This point is relevant because half of the respondents have some neurocognitive impairment. Considering that only 16.9% of the 478 total cases per NBS responded to the survey, this could be much higher, assuming that the respondents were those in the best condition. This should be discussed and considered a weakness.

Author Response

Thank you very much for taking the time to review our manuscript, “Expanded Newborn Screening for Inborn Errors of Metabolism in Louisiana (2005–2024): Outcomes.” We have carefully considered the reviewers’ comments and greatly appreciate their time, effort, and the many helpful suggestions and thoughtful critiques.

Below, we have addressed each comment individually. The original comment is italicized, our response follows “>>>”, and any text added to the manuscript is highlighted in red.

Please note that the current manuscript length is 11 pages, including 2 tables.

We look forward to your feedback and further guidance.

Reviewer 3

This work provides valuable information on the results of newborn screening in a specific location.

Abstract is OK

Keywords: words associated with the text should be included; suggestions include: neurodevelopmental outcomes, early onset, MCADD, newborn screening...

>>> We agreed and have added these keywords: newborn screening, clinical outcome, neurodevelopmental outcome, inborn error of metabolism, tandem mass spectrometry

Introduction is OK

Materials and Methods

Line 75: should explain why these pathologies are not included

>>> Clinical outcomes, including morbidity, mortality, and developmental status, were specifically analyzed for patients with inborn errors of metabolism (IEMs) detected by MS/MS. Certain metabolic conditions were excluded from this analysis for the following reasons:

  • Biotinidase deficiency, galactosemia, and carnitine uptake defect (CUD): Incomplete medical records or patients were not followed at our center, mention in the method, page 2, line 84-87.
  • Hemoglobinopathies, SCID, SMA, CAH, and congenital hypothyroidism: Our metabolic center does not follow patients with these conditions, so outcome data were not available.
  • Pompe disease and MPS I: Outcomes for these disorders have been extensively reported by multiple centers, and including them here would require additional data collection beyond the scope of this manuscript.

We have provided the reason for excluding these diseases, page 3, line 94-99

"Patients with hemoglobinopathies, SCID, SMA, CAH, and congenital hypothyroidism were not included in this analysis because our metabolic center does not follow patients with these conditions, and therefore outcome data were not available. Similarly, Pompe disease and MPS I were not included, as outcomes for these disorders have been extensively reported by multiple centers, and incorporating them here would require additional data collection beyond the scope of this manuscript."

Line 84: add the phrase: "Certain metabolic conditions were excluded from this analysis: biotinidase..."

>>> We have made change to this sentence per reviewer’s recommendation

Results

Line 177: should specify that of the total cases (81), 64 attended school.

>>> We have changed this sentence to “Of the total 81 respondents, 64 were attending school. Among these 64 patients…….” Page 6, line 186

Discussion

Line 215: These pathologies that were not found after 19 years. Are they common in the USA, or is it a genetically distinct population? This should be discussed, as it is striking that so many pathologies have gone undiagnosed after 19 years.

>>> Thank you for this comment. We would like to clarify that the paragraph refers to conditions not detected by the Louisiana newborn screening program during the study period (2005–2024), and does not imply that these diseases went undiagnosed in the population overall. The disorders in question are very rare, and their absence in our cohort is consistent with expected incidence rates in the United States rather than reflecting a genetically distinct population. We have revised the manuscript to clarify this point and provide context for the rarity of these condition, page 7, line 229-232 and provided the references to support the rarity of these diseases.

It is important to note that these conditions are extremely rare, and their absence in our cohort is consistent with their low expected incidence in the United States [Linder M, et al., 2010; Sarafoglou K, et al., 2011; Catsburg c, et al., 2022; Bayzaei Z, et al., 2024].”

Line 260: This point is relevant because half of the respondents have some neurocognitive impairment. Considering that only 16.9% of the 478 total cases per NBS responded to the survey, this could be much higher, assuming that the respondents were those in the best condition. This should be discussed and considered a weakness.

>>> Thank you for this comment. We agree that the low response rate raises the possibility of response bias. As noted in our Discussion, only 81 of 221 invited families responded, which may overrepresent those who are more engaged with medical care. Importantly, the questionnaire was completed by parents or caregivers, the majority of whom were healthy adults, rather than by the patients themselves.

We have already addressed the limitation of the survey in the final paragraph of the Discussion, noting that only 81 of 221 invited participants responded: Page 8, line 299-303

…the survey component is subject to responder bias, as only 81 of 221 invited participants responded, potentially overrepresenting patients with better outcomes or higher engagement with care. This may limit the generalizability of the neurodevelopmental and educational outcome data.